# Conceptual Framework for Trustworthy Artificial Intelligence: Combining Large Language Models with Formal Logic Systems

**Andrey Nechesov, Dmitry Kondratyev, Dmitry Sviridenko, Igor Anureev,**
**Natalia Garanina, Andrei Gumirow, Ivan Gorobets & Yana Dementyeva**
The Artificial Intelligence Research Center of Novosibirsk State University
1, Pirogova str., Novosibirsk, 630090, Russia
`nechesoff@gmail.com, d.kondratev2@g.nsu.ru,`
`d.sviridenko@nsu.ru, i.anureev@g.nsu.ru,`
`n.garanina@g.nsu.ru, a.gumirov@g.nsu.ru`
`i.gorobets@g.nsu.ru, y.dementeva@g.nsu.ru`

## Abstract

The paper explores the problem of building trustworthy artificial intelligence based on large language models and p-computable checkers. For this purpose we present a concept of framework for reliable verification of answers obtained by large language models (LLMs). We focus on the application of this framework to digital twin systems, particularly for smart cities, where LLMs are not yet widely used due to their resource intensity and potential for hallucination. Taking into account the fact that solution verification from a suitable set of tasks is p-computable and in most cases less complex than computing and implementing the whole task, we present a methodology that uses checkers to assess the validity of LLM-generated solutions. These checkers are implemented within the methodology of polynomial-time programming in Turing-complete languages, and guarantee a polynomial-time complexity. Our system was tested on the 2-SAT problem. This framework offers a scalable way to implement trustworthy AI systems with guaranteed polynomial complexity, ensuring error detection and preventing system hangups.

## 1 Introduction

Today, an increasing number of users are joining the use of artificial intelligence. Artificial intelligence is being applied to a wide range of tasks and this range is growing. However, there have already been many cases where the involvement of AI, in particular large language models, has led to incorrect and even dangerous decisions. Therefore, it is important for us to obtain a trusted artificial intelligence. In this paper, we propose a concept of framework for reliable verification of decisions obtained using a large language model. The topic of trustworthiness of artificial intelligence systems has recently received increasing attention from both AI researchers and AI systems users. It is the concept of designing and operating AI systems that are guaranteed to have the characteristics that we would normally ascribe to some agent. In this case, we usually talk about safety, security, responsibility, reliability, reproducibility, efficiency, productivity, transparency, confidentiality, fairness, ethics of its actions and results. All of this also applies to AI systems. The main challenge of trustworthy AI is to find an answer to the question, how to achieve it? And often the problem is to find a toolkit that will be used to achieve the desired result.

In the case of the symbolic approach, trust can be guaranteed by the presence of a reasonable, explainable, transparent and reliable development and operation environment based on logical and probabilistic principles. At the same time the use of machine learning technologies is dangerous due to high uncertainty and low level of transparency and validity of the results obtained. It is especially peculiar to the technology of artificial neural networks, in particular, to the technology of large language models (LLM), when we face the "black box" effect and hallucinations. At the moment, trusted artificial intelligence is most in demand for implementation in digital twin systems

(Goncharov & Nechesov (2023), Nechesov et al. (2025)). We are considering the construction of digital twins for smart cities (Nechesov & Ruponen (2024)), but we are not yet able to involve LLMs due to their unreliability and high resource intensity. However, there are still plenty of tasks that require involving LLMs to get at least inaccurate solutions for multiparametric problems that are impossible or difficult to solve using analytical methods based on available resources. We realize that it is necessary to check the correctness of these solutions. Therefore our concept is invented.

It is well known that solving NP-class problems (Prates et al. (2019)) has a high computational complexity above polynomial. However, verifying a solution is a more simple task and often belongs to class P and requires polynomial time. This concept allows us to reduce the cost of computational resources on the side of the digital twin due to verification of solutions by a checker working in polynomial time. The checkers themselves are written within the framework of polynomial-time programming methodology in Turing-complete languages (Goncharov et al. (2024)). Our methodology allows us to check whether our checker corresponds to class P. The polynomial complexity check is performed following the methodology of semantic programming, where tasks are formulated following to the task approach (Nechesov (2023)).

Let us note another advantage of our methodology. There are a lot of cases when implementation of task solution checker is simpler than implementation of solver for this task. Since our methodology is based on implementation of checkers instead of solvers, our approach allows reducing developer efforts in these cases.

The system, which is described in detail below, was tested on the 2-SAT problem. Scenarios were considered when the LLM gave the correct answers and when it was wrong. A finite set of checkers with partial order applied to evaluate the answers. Next, the jointness of the solver obtained using LLM and a certain checker from the set is evaluated. A solution can be decided as trusted only when the domain of the original problem coincides with the domain of the solver. The advantage of this framework is that it allows us to use trustworthy artificial intelligence systems while guaranteeing polynomial complexity. In addition, the system signals errors and hangups. The 2-SAT problem is only an illustrative example of understanding the concept of a conceptual environment for developing trusted intelligent systems. The concept itself is applicable to other classes of tasks.

This paper is divided into the following sections. The essence of the proposed concept and its fundamental foundations are outlined in the second section. The next section describes the testing of the system. A summary of the results is presented below in the conclusion.

## 2 FRAMEWORK

In this section we present a conceptual framework for ensuring trustworthiness of AI-based solvers (for example, LLMs).

First of all, we formalize the concepts of AI-based solver and checking environment.

Let letters $I$ and $O$ (possibly with indices and primes) denote sets.

An AI-based solver $S$ for inputs set $I$ and outputs set $O$ is a (possibly partial) function from $D \to R$. The partiality property means that there are inputs where the solver cannot solve the problem.

Let $bool = \{true, false\}$, and $dom(f)$ denotes the domain of a partial function $f$.

A checking environment $E$ is a pair $(\Psi, \prec)$ where $\Psi = \{\psi_1, \ldots, \psi_n\}$ is a finite set of partial functions $\psi_j \in D_j \times R_j \to bool$ called checkers, and $\prec$ is a partial order relation on $\Psi$ such that for all $1 \leq j, k \leq n$ the following properties hold for each input $i \in I$:

- $j < k \wedge \psi_j \prec \psi_k \wedge i \in dom(\psi_j) \Rightarrow i \in dom(\psi_k) \wedge \psi_k(i) = \psi_j(i)$;

- $\neg(\psi_j \prec \psi_k) \wedge \neg(\psi_k \prec \psi_j) \wedge i \in dom(\psi_j) \Rightarrow$
  $i \in dom(\psi_k) \wedge \psi_k(i) = \psi_j(i) \vee i \notin dom(\psi_k)$.

The checking environment $E$ is attached to the AI-based solver $S$, providing verification of the correctness of the output $o$ of $S$ at the input $i$, i.e. the correctness of the pair $(i, o)$, by applying checkers from $\Psi$. The $true$ value of the checker means that the solver returned the correct result $o$

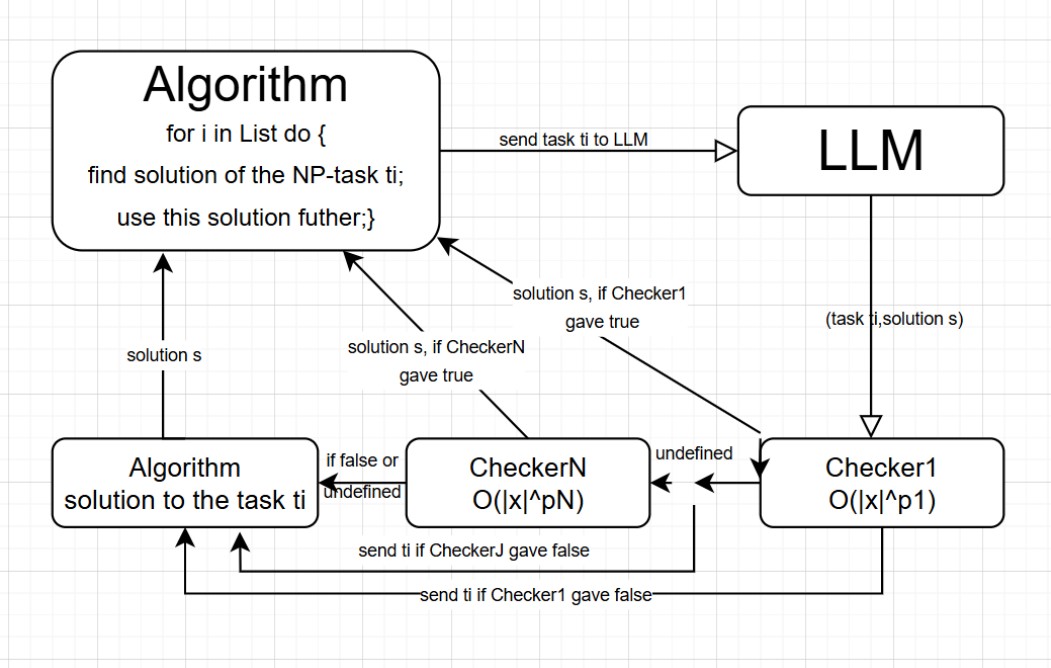

Figure 1: The process of solving problems in combination of algorithms and LLMs

at input $i$, the $false$ value does that the solver's result is incorrect at this input, and the undefined value $\perp$ does that the checker could not make an estimate.

The $\prec$ relation specifies the order in which the checkers are run.

If $\psi_j \prec \psi_k$, then the checker $\psi_k$ can run only after the checker $\psi_j$ returns the result, and whether the checker $\psi_k$ is started depends on this result. The checker $\psi_k$ is run only if the checker $\psi_j$ failed to check the pair $(i, o)$, i. e. $\psi_j$ returned an undefined value $\perp$. The first property ensures that if the checker $\psi_j$ did not fail to check the pair, then running the checker $\psi_k$ returns the same result, that is, it makes no sense to run it.

If $\neg(\psi_j \prec \psi_k)$, and $\neg(\psi_j \prec \psi_k)$, then the checkers $\psi_j$ and $\psi_k$ can be run in parallel. The second property ensures consistency of the results of parallel checkers, which means that it cannot happen that one of them returned $false$ and the other returned $true$.

A solver $S$ and an environment $E$ are consistent if $D_j \subseteq D$, and $E_j \subseteq E$ for each $1 \leq j \leq n$. The consistency property means that checkers work within the state space $D \times E$ of the solver.

Now we can define trustworthiness condition for AI-based solvers.

A solver $S$ is trustworthy w.r.t. an environment $E$ on a set $I' \subseteq I$ if for each $i \in I'$ there exists a path $\psi_{m_1} \prec \ldots \prec \psi_{m_l}$ such that $\psi_{m_r}(i) = \perp$ for each $1 \leq r \leq l - 1$, and $\psi_{m_l} \in bool$. The existence of the path means that checking environment estimates the solver on any inputs from $I'$.

A solver $S$ is trustworthy w.r.t. an environment $E$ if A solver $S$ is trustworthy w.r.t. an environment $E$ on a set $I$. This property means that solver $S$ is trustworthy at any inputs.

Let us note that the system $(S, E)$ consisting of a solver $S$ and checking environment $E$ can be considered as a hybrid intellectual system with intellectual part $S$ and analytical part $E$ that provides total or partial (on a subset of inputs) trustworthiness of $S$. Since checker implementation is simpler than solver implementation in many practical cases, hybrid nature of our approach allows reducing developer efforts in practice.

There are two problems that arise when building such a hybrid AI.

First, we need to make sure that all checkers from the checking environment are working correctly. To address this problem we use formal verification methods (in particular, the deductive verification method Hähnle & Huisman (2019)) for ensuring the correctness of the checkers themselves).

Second, we need to ensure relatively small time complexity of the checkers included in the checking environment. To address this problem we use polynomial-time programming methodology in Turing-complete languages (Goncharov et al. (2024)) which allows us to check whether a checker corresponds to class P.

With this in mind, we define a conceptual framework as the quadruple $(S, E, \Delta_1, \Delta_2)$, where $(S, E)$ has already been defined above, $\Delta_1$ is a set of checkers verification tools, and $\Delta_2$ is a set of checkers complexity assessment tools.

## 3    EXPERIMENTS

In this section, we illustrate the main components of the conceptual framework using the example of solving 2-SAT problems.

### 3.1    APPLICATION OF OUR FRAMEWORK TO 2-SAT PROBLEM

2SAT is an P-class problem of assigning values to binary variables to perform a conjunction of $k$ disjunctions. It is a special case of the general Boolean satisfiability problem. Thus, Problem of 2-SAT can be stated as: Given Conjunctive Normal Form F with each clause having only 2 terms:

$$F = (A_1 \vee B_1) \wedge (A_2 \vee B_2) \wedge (A_3 \vee B_3) \wedge \cdots \wedge (A_m \vee B_m)$$

Is it possible to assign such values to the variables so that the Conjunctive Normal Form is TRUE?

The checking environment $E$ of our conceptual environment consists of two functions: `twosat_solver` and `sat_solution_checker`. Implementations of these functions are available in our repositories Kondratyev (2025a;b) and also in Appendix A and Appendix B. These implementations correspond to our polynomial programming methodology (The corresponding polynomiality tests make up the component $\Delta_2$).

The `twosat_solver` function allows us to check unsatisfiability case. Thus, $dom(\texttt{twosat\_solver}) = \{(i, o) \mid i \text{ is a 2-SAT task, and } o = unsat\}$. We define formal specifications of this function to verify its implementation. This function checks the property of existence of path from $x$ to $\neg x$ and existence of path from $\neg x$ to $x$ in implication graph for any formula variable $x$. This property is important in the field of 2-SAT problem. If this property holds, than formula is unsatisfiable. Let us note that implementation of whole 2-SAT solver requires not only implementing check of this property but also implementing search of all strongly connected components of implication graph and topological sort of these components to build solution of 2-SAT problem Aspvall et al. (1979). But our approach based on using checkers in our environments allows us to avoid implementing whole 2-SAT solver. Thus, we use the `twosat_solver` function instead of implementation of whole 2-SAT solver. Consequently, we have reduced developer efforts in this case. Another function `twosat-solver` used in specifications is implementation of solution of SAT problem using generation of all combinations of possible values of formula variables. Specifications describe equivalency between this ineffective simple implementation and polynomial complex implementation of 2-SAT solution algorithm in the case of unsatisfiable formula. We have used the C-lightVer deductive verification tool Kondratyev & Nepomniaschy (2022) (an element of $\Delta_1$ from the conceptual environment) to prove property of this equivalence described in specifications. This proof allows us to guarantee that implementation of this unsatifiablity checker is trustworthy Kondratyev et al. (2025).

The `sat_solution_checker` function allows us to check whether a set of variable values proposed by LLM is solution of 2-SAT problem. Thus, $dom(\texttt{sat\_solution\_checker}) = \{(i, o) \mid i \text{ is a 2-SAT task, and } o = sat\}$. The polynomial implementation of this function is based on iterations over disjunctions and variables of formula.

We use ChatGPT and DeepSeek as solvers in our conceptual framework and apply the prompt from Appendix C to solve 2-SAT problems on these LLMs.

Application of our framework to 2-SAT problem allows us to solve tasks that can be reduced to 2-SAT problem.

## 3.2 Heterogeneous resource allocation (HRA) in the case of two resources and one-level dependencies between tasks

We use the task of heterogeneous resource allocation (HRA) from the paper Ait Aba et al. (2019) as the first case study. We consider that we have two platforms, each with an unbounded number of processors. We want to execute an application represented as a Directed Acyclic Graph (DAG) using these two platforms. Each task of the application has two possible execution times, depending on the platform it is executed on. Finally, there is a cost to transfer data from one platform to another one between successive tasks. Maximum depth of this DAG is 1. So task can depend only on task that has no dependencies. Tasks could be executed in parallel. Each task predecessors could be ran on different platforms (i.e when task 1 depends on 2 and 3, these tasks may be executed on different platforms).

The goal is to calculate minimum possible time of total DAG execution and return platform for each task.

We use a polynomial algorithm for solving this task and apply obtained polynomial algorithm to this task Ait Aba et al. (2019). This algorithm is based on reduction of this task to 2-SAT problem. Reduction of considering task to 2-SAT is implemented by us and available in the repository Kondratyev (2025a). We use application of our framework to 2-SAT problem for solving tasks obtained by this reduction.

Two examples of our first case study lead to 15 2-CNF formulas: 8 2-SAT formulas resulted from first example and 7 2-CNF formulas resulted from second example. On the one hand, all 8 2-CNF formulas corresponded to first example have been solved by both LLMs (ChatGPT and DeepSeek). Our checkers allowed us to verify solutions of these 8 2-SAT cases. On the other hand, only one formula corresponding to second example has been solved by our solver. Let us note that this formula has been solved by DeepSeek and has not been solved by ChatGPT. The set of 7 2-CNF formulas corresponded to second example contain interesting case when DeepSeek reports about unsatisfiability but our unsatisfiability checker allows us to discover that this formula is not unsatisfiable. The representation of this formula from prompt in DIMACS CNF format is follow:

```
p cnf 3 6
-3 2 0
-1 2 0
-3 -2 0
1 2 0
-1 -2 0
3 2 0
```

Let us consider DIMACS cnf format which is used by a lot of modern SAT solvers and by us to define 2-SAT instances in prompt. The number of variables and the number of clauses are defined by the line `p cnf variables clauses`. Each of line below specifies a clause: a positive literal is denoted by the corresponding number, and a negative literal is denoted by the corresponding negative number. The last number in a line should be zero.

Let us consider the classic representation of this formula:

$$(\neg x_3 \vee x_2) \wedge (\neg x_1 \vee x_2) \wedge (\neg x_3 \vee \neg x_2) \wedge (x_1 \vee x_2) \wedge (\neg x_1 \vee \neg x_2) \wedge (x_3 \vee x_2)$$

This formula is not unsatisfiable (for example, this formula is true in the case of $x_1 = false$, $x_2 = true$ and $x_3 = false$).

This case is interesting since unsatisfiable case is more difficult to analyze due to absence of proposed solution in this case.

### 3.3 Applications in Smart Cities and Multi-Blockchains

The future of urban management is being redefined through the convergence of smart city initiatives, multi-blockchain architectures, and advanced artificial intelligence. This section presents an integrated framework where diverse blockchain systems store and process information across various domains of a smart city, smart contracts execute all business logic autonomously, and trustworthy AI underpins decision-making. Moreover, a polynomial complexity HRA-task is employed to maintain efficient control over the multi-blockchain ecosystem.

#### 3.3.1 Architecture of the Multi-Blockchain Ecosystem

The framework implements a three-levels hierarchy:

1. **Master Blockchain (1st Level)**:
   - Root layer with maximum decentralization/security (e.g., PoW/BFT)
   - Manages cross-domain coordination and integrity proofs
2. **Sector-Specific Blockchains (2nd Level)**:
   - Domain-optimized consensus (PoS for energy, PBFT for emergency services)
   - Interfaces between subsectors and master chain
3. **Subsector Blockchains (3rd Level)**:
   - High-throughput chains for granular operations (DAG-based consensus)
   - Examples: traffic light control, household energy metering

#### 3.3.2 Coordination and Processing Workflow

- **Bottom-Up Processing**:

$$3\text{rd level} \rightarrow 2\text{nd level} \rightarrow 1\text{st level}$$

  Parent chains process only after child chains complete
- **Resource Allocation**:
  - GPU-intensive: PoW-like consensuses
  - CPU-intensive: BFT/PoS consensuses
  - Single-device execution per blockchain

#### 3.3.3 Polynomial-Time Heterogeneous Resource Allocation (HRA) Task

Let:

$$\mathcal{B} = \{B_1, B_2, \ldots, B_n\} \text{ (blockchains)}$$
$$\mathcal{R} = \{\text{GPU}, \text{CPU}\}$$
$$T(B_i) : \text{Processing time for blockchain } B_i$$
$$B_j \prec B_k : \text{Processing order constraint}$$

**Objective**:

$$\min \left( \max_{r \in \mathcal{R}} \left( \sum_{B_i \text{ assigned to } r} T(B_i) \right) \right)$$

**Constraints**:

- Order preservation:

$$\forall B_j \prec B_k, \text{ start\_time}(B_j) + T(B_j) \leq \text{start\_time}(B_k)$$

- Consensus-specific resource assignment:

$$\text{GPU-required } B_i \Rightarrow B_i \text{ assigned to GPU device}$$

Since the master blockchain will be processed after we have processed all other blockchains, this problem is reduced to the HRA problem for a DAG- graph with two layers. This problem can be solved in polynomial time!

The integration of multi-blockchain storage, smart contracts, trustworthy AI, and efficient resource allocation creates a resilient and adaptive infrastructure for smart cities. By partitioning data into specialized ledgers, automating business processes via smart contracts, and underpinning operations with transparent and accountable AI, urban systems can achieve unprecedented levels of efficiency, security, and scalability. Furthermore, the deployment of a polynomial complexity heterogeneous resource allocation algorithm provides the necessary control to manage diverse and dynamic resources across the entire multi-blockchain ecosystem.

### 3.4  WIRELESS SENSOR NETWORK (WSN) CONNECTIVITY ANALYSIS

We use the task of wireless sensor network connectivity analysis from the paper Biró & Kusper (2018) as the another one case study. Since wireless sensor networks (WSN) are widely applied to such perspective area as smart city creation Belghith & Obaidat (2016), considering task is important Faye & Chaudet (2015).

This task is based on representation of communications between sensors as graph. This graph is referred to as communication graph. Vertexes of this graph are sensors. There is edge between sensors if and only if direct communication between these sensors exists. Let us note that this graph is directed due to possibility of only one-side direct communication in some cases (for example, when distance between two sensors does not allow sending messages from sensor with less powerful transmitter to sensor with more powerful transmitter but allows sending messages from sensor with more powerful transmitter to sensor with less powerful transmitter). Considering task is to check whether communication graph satisfies the following property: ability of each sensor to communicate with each another sensor with opportunity of using other sensors as repeaters. This property is equivalent of strong connectivity of communication graph.

The reduction of question about strong connectivity of communication graph to black-and-white 2-SAT problem has been described in the paper Biró & Kusper (2018). This reduction is interesting due its simplicity: each edge $(a, b)$ is translated to implication $a \rightarrow b$ ($\neg a \vee b$ conjunct in obtained 2-cnf formula). But authors of the paper Biró & Kusper (2018) have proved that it is necessary to avoid cases when all variables (vertexes) have $true$ values in 2-SAT solution and when all variables (vertexes) have $false$ values in 2-SAT solution. Authors of the paper Biró & Kusper (2018) have proposed to add the following two conjunctions to the formula to solve this problem: $x_1 \vee x_2 \vee \ldots \vee x_{n-1} \vee x_n$ and $\neg x_1 \vee \neg x_2 \vee \ldots \vee \neg x_{n-1} \vee \neg x_n$ (where $x_1, x_2, \ldots, x_{n-1}, x_n$ denotes all formula variables (all graph vertexes)). 2-SAT problem with these two additional constraints is referred to as black-and-white 2-SAT problem.

Authors of the paper Biró & Kusper (2018) have proved that communication graph is strongly connected if and only if corresponding black-and-white 2-SAT problem is unsatisfiable. But it is necessary to reduce considering task to ordinary 2-SAT problem instead of black-and-white 2-SAT problem to solve obtained 2-cnf formula in polynomial time. Thus, we propose reduction of considering task to ordinary 2-sat problem. We state that two additional constraints can be replaced by statement of presence of pair of variables that have different values in obtained solution. We suggest to check this statement in the iteration over all pair of variables. We propose to add to the formula constraint $(x_i \vee x_j) \wedge (\neg x_i \vee \neg x_j)$ (where $x_i$ and $x_j$ are iterated variables) on each iteration instead of adding big "black-and-white" constraints. We try to solve obtained 2-SAT problem on each iteration using application of our framework to 2-SAT problem. If obtained formula occurs satisfiable on some iteration then communication graph is not strongly connected. Else if obtained formula is unsatisfiable on each iteration then communication graph is strongly connected. Thus we improve result from the paper Biró & Kusper (2018) by reducing question of graph strong connectivity to ordinary 2-SAT problem instead of black-and-white 2-SAT problem. We have implemented this reduction in our repository Kondratyev (2025b).

Let us consider example of wireless sensor network described in the paper Biró & Kusper (2018). Corresponding communication graph is presented on Figure 2.

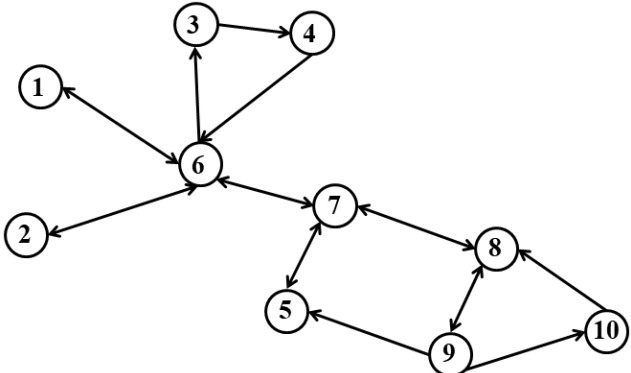

Figure 2: Communication graph of wireless sensor network described in the paper Biró & Kusper (2018)

Application of our approach to this communication graph results to 45 2-cnf formulas due to count of possible pairs of 10 variables. All of these 45 formulas are unsatisfiable. Thus, this communication graph is strongly connected. For example, let us consider the representation in DIMACS CNF format of the first formula from these 45 formulas:

```
p cnf 10 20
-1 6 0
-6 1 0
-2 6 0
-6 2 0
-6 3 0
-3 4 0
-4 6 0
-8 7 0
-6 7 0
-7 6 0
-5 7 0
-7 5 0
-7 8 0
-8 9 0
-9 8 0
-9 10 0
-10 8 0
-9 5 0
1 2 0
-1 -2 0
```

The classic representation of this formula has the following form:

$$(\neg x_1 \lor x_6) \land (\neg x_6 \lor x_1) \land (\neg x_2 \lor x_6) \land (\neg x_6 \lor x_2) \land (\neg x_6 \lor x_3) \land (\neg x_3 \lor x_4) \land (\neg x_4 \lor x_6) \land (\neg x_8 \lor x_7) \land (\neg x_6 \lor x_7) \land (\neg x_7 \lor x_6) \land (\neg x_5 \lor x_7) \land (\neg x_7 \lor x_5) \land (\neg x_7 \lor x_8) \land (\neg x_8 \lor x_9) \land (\neg x_9 \lor x_8) \land (\neg x_9 \lor x_{10}) \land (\neg x_{10} \lor x_8) \land (\neg x_9 \lor x_5) \land (x_1 \lor x_2) \land (\neg x_1 \lor \neg x_2)$$

The first 18 conjuncts of this formula correspond to connection graph edges. The last 2 conjuncts of this formula are resulted of first iteration over variables pairs. These conjuncts state that the values of $x_1$ and $x_2$ variables are different.

Applications of solvers from our framework to this formula led to the opposite results. On the one hand, the direct answer of DeepSeek is that this formula is satisfiable. Our unsatisfiability checker shows that it is wrong answer. On the other hand, DeepSeek generates Python code to solve 2-SAT tasks. Execution of this code results to correct answer about unsatisfiability of this formula. Let us

note that application of ChatGPT to this formula lead to correct answer about unsatisfiability of this formula. This case demonstrates importance of using our unsatisfiability checker.

Question of connectivity robustness of wireless sensor network relative to removing nodes or edges is important Dagdeviren & Akram (2019). Removing direct connection from sensor 7 to sensor 6 results to modification of the example presented at Figure 3.

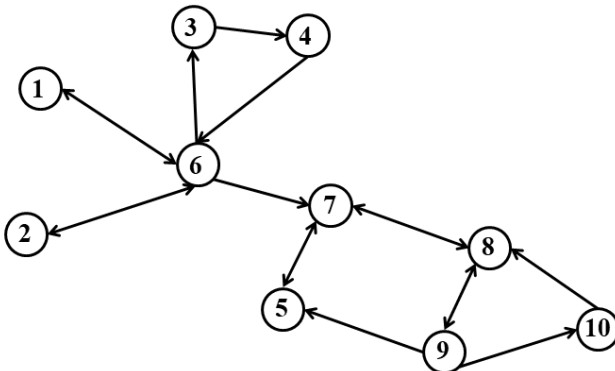

Figure 3: Communication graph of wireless sensor network described in the paper Biró & Kusper (2018) without the direct communication from sensor 7 to sensor 6

Application of our approach to this modified communication graph results to only 4 2-cnf formulas since solution has been found on fourth iteration. Let us consider this satisfiable formula obtained on fourth iteration:

```
p cnf 10 19
-1 6 0
-6 1 0
-2 6 0
-6 2 0
-6 3 0
-3 4 0
-4 6 0
-8 7 0
-6 7 0
-5 7 0
-7 5 0
-7 8 0
-8 9 0
-9 8 0
-9 10 0
-10 8 0
-9 5 0
1 5 0
-1 -5 0
```

The classic representation of this formula has the following form:

$$(\neg x_1 \vee x_6) \wedge (\neg x_6 \vee x_1) \wedge (\neg x_2 \vee x_6) \wedge (\neg x_6 \vee x_2) \wedge (\neg x_6 \vee x_3) \wedge (\neg x_3 \vee x_4) \wedge (\neg x_4 \vee x_6) \wedge$$
$$(\neg x_8 \vee x_7) \wedge (\neg x_6 \vee x_7) \wedge (\neg x_5 \vee x_7) \wedge (\neg x_7 \vee x_5) \wedge (\neg x_7 \vee x_8) \wedge (\neg x_8 \vee x_9) \wedge (\neg x_9 \vee x_8) \wedge$$
$$(\neg x_9 \vee x_{10}) \wedge (\neg x_{10} \vee x_8) \wedge (\neg x_9 \vee x_5) \wedge (x_1 \vee x_5) \wedge (\neg x_1 \vee \neg x_5)$$

The first 17 conjuncts of this formula correspond to connection graph edges. The last 2 conjuncts of this formula are resulted from fourth iteration over variables pairs. These conjuncts state that the values of $x_1$ and $x_5$ variables are different.

Execution of Python code generated by ChatGPT for solving this task has crashed with runtime error. Thus, we have undefined result of ChatGPT solver in this case. On the one hand, the direct

answer of DeepSeek is statement about satifiable of this formula. But our solution checker shows that solution proposed by DeepSeek in direct answer is incorrect. On the other hand, execution of Python code generated by DeepSeek results to correct solution. This correct solution is the following assignment: $x_1 = false, x_2 = false, x_3 = false, x_4 = false, x_5 = true, x_6 = false, x_7 = true, x_8 = true, x_9 = true, x_{10} = true$. Let us note that we do not need in using big "black-and-white" constraints to achieve this result. This case demonstrates importance of using our solution checker.

## 4    CONCLUSION

This paper presents a novel framework for the reliable verification of answers obtained by large language models (LLMs), with a focus on their application in digital twin systems for smart cities. Our experiments, conducted using the 2-SAT problem, demonstrated the effectiveness of the framework in correctly identifying trusted solutions, even in the presence of incorrect or suboptimal responses from the LLM. Let us note that advantages of reduction of tasks to 2-SAT led us to such improvement of result of the paper Biró & Kusper (2018) as reduction of question about graph strong connectivity to ordinary 2-SAT problem instead of black-and-white 2-SAT problem.

The mathematical foundation of the concept, coupled with successful experimental results, provides strong evidence for the feasibility of using this framework to integrate trustworthy AI into resource-constrained environments, such as digital twins. This approach not only guarantees polynomial-time complexity but also offers a robust mechanism for error detection and system stability. Moreover our methodology can reduce developer effort due to simplicity of checker implementation relative to solver implementation.

Future work could focus on extending the framework to other problems, optimizing the verification process, and exploring its application in real-world smart city implementations. Overall, this research contributes a significant step toward realizing reliable and efficient AI systems, facilitating their safe deployment in complex, critical environments.

### ACKNOWLEDGMENTS

This work was supported by a grant for research centers, provided by the Analytical Center for the Government of the Russian Federation in accordance with the subsidy agreement (agreement identifier 000000D730324P540002) and the agreement with the Novosibirsk State University dated December 27, 2023 No. 70-2023-001318.

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

## A   APPENDIX A

We implement checking unsatisfiability as following `twosat_solver` function written in C programming language (formal specifications of this function are written in C comments using Applicative Common Lisp language):

```
/*
(and
    (integerp
        variable_count
    )
    (<
        0
        variable_count
    )
    (integer-listp
        implication_graph_transitive_closure
    )
    (=
        (len
```

```
                    implication_graph_transitive_closure
            )
            (*
                4
                (*
                    variable_count
                    variable_count
                )
            )
        )
)
*/
int twosat_solver(int variable_count,
    int implication_graph_transitive_closure[])
{
    int x = 0;
    int satisfiable = 1;
    /*
    (and
        (integerp
            variable_count
        )
        (<
            0
            variable_count
        )
        (integer-listp
            implication_graph_transitive_closure
        )
        (=
            (len
                implication_graph_transitive_closure
            )
            (*
                4
                (*
                    variable_count
                    variable_count
                )
            )
        )
        (integerp
            x
        )
  (<=
            0
            x
        )
        (implies
            (=
                satisfiable
                0
            )
            (and
                (=
                    (nth
                        (+
                            (*
                                x
```

```
                                        (*
                                            variable_count
                                            2
                                        )
                                    )
                                    (+
                                        x
                                        variable_count
                                    )
                                )
                                implication_graph_transitive_closure
                            )
                            1
                        )
                        (=
                            (nth
                                (+
                                    (*
                                        (+
                                            x
                                            variable_count
                                        )
                                        (*
                                            variable_count
                                            2
                                        )
                                    )
                                    x
                                )
                                implication_graph_transitive_closure
                            )
                            1
                        )
                        (<
                            x
                            variable_count
                        )
                    )
                )
            )
    */
    while (x < variable_count && satisfiable == 1)
    {
        if (implication_graph_transitive_closure[
                x + variable_count + x * 2 * variable_count] == 1 &&
            implication_graph_transitive_closure[
              x * 2 * variable_count + x +
                variable_count * 2 * variable_count] == 1)
        {
            satisfiable = 0;
        }
        else
        {
            x++;
        }
    }
    return satisfiable;
}
/*
```

```
(implies
    (=
        satisfiable
        0
    )
    (not
        (twosat-solver
            (boolean-variable-values
                variable_count
                nil
            )
            variable_count
            implication_graph_transitive_closure
        )
    )
)
*/
```

## B  APPENDIX B

We implement checking solution of 2-SAT problem as following sat_solution_checker function written in C programming language:

```
int sat_solution_checker(int variable_count,
                         int disjunction_count,
                         int* twocnf_formula,
                         int* variable_values)
{
    int result = 1;
    for (int i = 0; (i < disjunction_count) && (result == 1);
         i++)
    {
        int index_i = 2*i;
        int first_literal = twocnf_formula[index_i];
        int second_literal = twocnf_formula[index_i + 1];

        int first_literal_variable;
        if (first_literal > 0)
        {
            first_literal_variable = first_literal;
        }
        else
        {
            first_literal_variable = -first_literal;
        }

        int first_variable_value;
        int first_variable_find = 0;

        for (int i = 0;
             (i < variable_count) && (first_variable_find == 0);
             i++)
        {
            int variable;
            if (variable_values[i] > 0)
            {
                variable = variable_values[i];
```

```
        }
        else
        {
            variable = -variable_values[i];
        }
        if (variable == first_literal_variable)
        {
            if (variable_values[i] > 0)
            {
                first_variable_value = 1;
            }
            else
            {
                first_variable_value = 0;
            }
            first_variable_find = 1;
        }
    }

    int second_literal_variable;
    if (second_literal > 0)
    {
        second_literal_variable = second_literal;
    }
    else
    {
        second_literal_variable = -second_literal;
    }

    int second_variable_value;
    int second_variable_find = 0;

    for (int i = 0;
         (i < variable_count) && (second_variable_find == 0);
         i++)
    {
        int variable;
        if (variable_values[i] > 0)
        {
            variable = variable_values[i];
        }
        else
        {
            variable = -variable_values[i];
        }
        if (variable == second_literal_variable)
        {
            if (variable_values[i] > 0)
            {
                second_variable_value = 1;
            }
            else
            {
                second_variable_value = 0;
            }
            second_variable_find = 1;
        }
    }
```

```
        if ((first_literal > 0) && (second_literal > 0))
        {
            if ((first_variable_value == 0) &&
                (second_variable_value == 0))
            {
                result = 0;
            }
        }
        else if ((first_literal > 0) && (second_literal < 0))
        {
            if ((first_variable_value == 0) &&
                (second_variable_value == 1))
            {
                result = 0;
            }
        }
        else if ((first_literal < 0) && (second_literal > 0))
        {
            if ((first_variable_value == 1) &&
                (second_variable_value == 0))
            {
                result = 0;
            }
        }
        else
        {
            if ((first_variable_value == 1) &&
                (second_variable_value == 1))
            {
                result = 0;
            }
        }
    }
    return result;
}
```

## C   APPENDIX C

We use the following prompt to solve 2-SAT problems on LLMs:

```
We consider 2-satisfiability (2-SAT) problem. In computer science,
2-SAT is a computational problem of assigning values to variables,
each of which has two possible values, in order to satisfy
a system of constraints on pairs of variables. It is a special
case of the general Boolean satisfiability problem. Instances of
the 2-satisfiability problem are typically expressed as Boolean
formulas of a special type, called 2-conjunctive normal form
(2-CNF) formulas. A 2-satisfiability problem may be described
using a Boolean expression with a special restricted form.
It is a conjunction (a Boolean and operation) of clauses, where
each clause is a disjunction (a Boolean or operation) of two
variables or negated variables. The variables or their negations
appearing in this formula are known as literals.

The goal is to solve a 2-SAT problem on input Boolean formula.

Use a polynomial algorithm for solving this task.
```

Input data is Boolean formula encoded in DIMACS CNF format.

Input data format is:
p cnf <number of variables 'N'> <number of clauses 'S'>
After that S lines of:
<first literal i where i is variable number in the case of positive literal or i is negative variable number in the case of negative literal> <second literal j where j is variable number in the case of positive literal or j is negative variable number in the case of negative literal> 0

Output data format is solution encoded in DIMACS CNF format.

Output data format is:
s <result k where k is "SATISFIABLE" in the case of satisfiable input formula or k is "UNSATISFIABLE" in the case of unsatisfiable input formula>
If input formula is satisfiable then representation of variable assignments in 1 line of:
v <variable assignments a_i separated by space (where i is variable number i from 1 to N and a_i is i in the case of assignment of true to variable i or a_i is -i in the case of assignment of false to variable i)> 0

Solve this task for this example:
...

