# OpenReview forum: "Conceptual Framework for Trustworthy Artificial Intelligence: Combining Large Language Models with Formal Logic Systems"
_mathai.club/MathAI/2025/Conference — MathAI 2025 Oral_

### Official Review · Reviewer_GAnV · 2025-02-25
**The topic is relevant, but the presentation of the material is rather chaotic.**

**Rating:** 7
**Confidence:** 4

**Review:**

The paper proposes a conceptual idea for verifying the results of an LLM agent using specialized checker functions. The purpose of these checkers is to ensure that the LLM's output is correct within polynomial time (p-class problem). It is stated that this framework can be used for digital twins of smart cities. However, this is mentioned only in the introduction and conclusions, while the main text of the paper does not elaborate on this, making the connection unclear. There is a reference to a problem statement in another article, but it is not clear what exactly is written in that article, as no relevant information is provided in the current paper.

A separate section is dedicated to the experiment, but the experiment is not described properly, making it unclear what exactly was done and how it was carried out.

Overall, the paper appears to be in a very raw state. There might indeed be an interesting result, but it is not effectively communicated to the reader. Additionally, there are several specific issues with the presentation of the material:

line 34, 218:  The expression large langauge model is not abbreviated, although the abbreviation was previously introduced

line 39: "to transform imprecise queries into clear tasks" is not good explanation. Why we change query to task and how? There aren't  any examples and explanation.

line 43:  It is stated that " verifying a solution is a task that belongs to class P and requires polynomial time." Why is it TRUE?

line 51: There are no explanation what is 2-SAT task

line 74: There is "is a pair ($\Psi,\prec)$ ", but what is $\prec$ is not explain

lines 72-81 : $ \psi -> bool$, bool = {True, False}, but line 80 $\psi = \perp$ , $\perp$ is not in bool. It is a mistake if you try to use mathematical language.

lines 137-138: "We have used the C-lightVer deductive verification tool Kondratyev & Nepomniaschy (2022) to prove property of this equivalence described in specifications." This information is only link without explanation.

line 218, 229: THe connection with smart city is not obvious

---

### Official Review · Reviewer_oW6q · 2025-02-25
**CONCEPTUAL FRAMEWORK FOR TRUSTWORTHY ARTIFICIAL INTELLIGENCE: COMBINING LARGE LANGUAGE MODELS WITH FORMAL LOGIC SYSTEMS**

**Rating:** 5
**Confidence:** 4

**Review:**

The paper explores the problem of building trustworthy artificial intelligence based on large language models and p-computable checkers.  In the second section, the author presented a conceptual framework for checking the correctness of outputs of AI-based
solvers. Wheres in the third section is devoted to the experiments by taking the task of heterogeneous resource allocation as the case study.

The topic of the paper is very interesting and important. However, the paper lacks detailed results from both theoretical and practical perspectives. In the abstract, it is mentioned that the presented framework can be used for digital twins of smart cities while the main text of the paper does not clarify this.

---

### Official Review · Reviewer_DKHh · 2025-02-26
**Review of Paper 22**

**Rating:** 6
**Confidence:** 3

**Review:**

**Overall Review**

This paper proposes a framework for building trustworthy artificial intelligence by combining large language models with formal logic systems. The methodology involves using polynomial-time checkers to verify the correctness of solutions generated by LLMs, ensuring reliable and efficient AI systems.

**Strength**

•	The objective of the paper to put emphasis on building trustworthy AI is crucial, especially given the potential for hallucinations and errors in LLM-generated outputs.

**Weakness**

•	The experiments are focused solely on 2-SAT problems, hence it appears that scope of experimentation was limited.

•	While the paper discusses the relevance of the framework to smart cities, it does not include extensive real-world examples or case studies to illustrate its application in such environments.

•	Noticed a typographical error in Line 128-129: "If this property holds than formula is unsatisfiable." should be "If this property holds, then the formula is unsatisfiable."

Providing case studies of the framework's application in smart cities would enhance the paper's impact. I am giving it a borderline score.

---

### Official Review · Reviewer_n5kn · 2025-02-26
**The relevant paper but needs more explanations on key points (Updated)**

**Rating:** 6
**Confidence:** 4

**Review:**

The paper presents the framework for validating answers of LLMs using the p-computable checkers. Authors states, that such framework can be useful for digital twins of smart cities. The idea of this paper is relevant and practically applicable, however the explanation sometimes is not complete. Here is the strengths and weaknesses of the paper.

**Strengths**

- The problem discussed in the paper is relevant and practically applicable;

- The proposed solution has its novelty both in science and practical domain.


**Weaknesses**

- The required size of paper was 6-9 pages of content without the references and Appendix, however the paper contains near 4.5 pages of content, where almost 0.5 is the prompt that could be moved to Appendix;

- The paper lacks of explanation of 2-SAT task;

-  Authors states the applicability of the proposed framework for digital twins of smart cities but it is not detailed inside the main content of the paper;

-  The connection between processes inside LLMs and logical systems are not explained;

Providing more explanations about practical applicability of the framework in real-world scenarios and the theoretical background of connection between LLMs and logical systems would enhance the paper's quality.

---

### Official Review · Reviewer_qdTe · 2025-02-26
**Conceptual Framework for Trustworthy Artificial Intelligence: Combining Large Language Models with Formal Logic Systems. Reviewer recommends to include it in the Program of the International conference “Mathematics of Artificial Intelligence” (24-28 March 2025, Sochi) with its publication.**

**Rating:** 9
**Confidence:** 4

**Review:**

The article is devoted to the problem of building trustworthy artificial intelligence based on large language models and p-computable checkers. The authors present a concept of framework for reliable verification of answers obtained by large language models (LLMs) for its application, first of all, to digital twin systems, particularly for smart cities, where the resource intensity is needed and dangerous of hallucination is high. The authors note that solution verification from a suitable set of tasks is p-computable and in most cases less complex than computing the whole task. Correspondingly, they put forward a methodology that uses checkers to assess the validity of LLM-generated solutions. These checkers are implemented within the methodology of polynomial-time programming in Turing-complete languages, and guarantee a polynomial-time complexity. It is especially valuable that the proposed system was successfully tested on the 2-SAT problem. The proposed framework offers a scalable way to implement trustworthy AI systems with guaranteed polynomial complexity, ensuring error detection and preventing system hangups.
  Of course, it is important to obtain a trusted artificial intelligence. In this article, the authors propose a concept of framework for reliable verification of decisions obtained using a large language model. The authors are considering the construction of digital twins for smart cities, but they are not yet able to involve LLMs due to their unreliability and high resource intensity. But there are plenty of tasks that require involving LLMs to transform imprecise queries into clear tasks that can be processed by the system. For its application, it is necessary to check the correctness of the solution. The authors invent a corresponding concept taking into account that solving NP-class problems has a high computational complexity above polynomial. But verifying a solution is a task that belongs to class P and requires polynomial time. The corresponding authors’ concept allows to reduce the cost of computational resources on the side of the digital twin due to verification of solutions by a checker working in polynomial
time. The checkers themselves are written within the framework of polynomial-time programming methodology in Turing-complete languages. The invented methodology allows to check whether our checker corresponds to class P. The polynomial complexity check is performed following the methodology of semantic programming, where tasks are formulated following to the task approach (Nechesov (2023)).
  The mathematical foundation of the concept, coupled with successful experimental results, provides strong evidence for the feasibility of using this framework to integrate trustworthy AI into resource-constrained environments, such as digital twins. The reviewer agrees with the authors that this approach not only guarantees polynomial-time complexity but also offers a robust mechanism for error detection and system stability. Future work could focus on extending the framework to other problems, optimizing the verification process, and exploring its application in real-world smart city implementations. Overall, this research contributes a significant step toward realizing reliable and efficient AI systems, facilitating their safe deployment in complex, critical environments.
     The list of references in the article includes 12 appropriate literary sources. The quality, clarity, originality and significance of this work are high.
      I recommend to include it in the Program of the International conference “Mathematics of Artificial Intelligence” (24-28 March 2025, Sochi) with its publication.

---

### Official Review · Reviewer_3Ebt · 2025-02-26
**Conceptual Framework for Trustworthy Artificial Intelligence: Combining Large Language Models with Formal Logic Systems**

**Rating:** 9
**Confidence:** 3

**Review:**

The expanding implementation of AI is increasingly exacerbating the issues of trust in the solutions obtained using it, especially in connection with the use of large language models (LLM). Traditional problems with errors in program code are becoming more acute (on the website of the Center for System Software Security Research opened in 2023 https://portal.linuxtesting.ru/ over 500 errors in the open source code of the Linux kernel have already been found and fixed over the course of a year), issues of explainability of solutions generated by AI, creation of reliable software for environments with limited resources (smart cities, digital twins) and other issues. The work is devoted to these urgent problems, the work proposes a new software system for testing solutions obtained using large language models.
It is gratifying that open source software continues to win compared to proprietary software - the work shows examples of applications of the developed system, in which DeepSeek coped with the test task, and ChatGPT did not. The proposed system is suitable for practical applications, as it has a polynomial complexity of the program code used.
Among the shortcomings of the article, we note the absence of a number of explanations (it is not clear what the "2-SAT task" consists of, etc.) while it is possible to increase the volume of the article, which is recommended to the author of the article when editing it based on the comments of reviewers, it is necessary to format the list of references, eliminating typos.
The article is of significant interest, both theoretically and for applications. The reviewer recommends the article for a report on it at the MathAI 2025 conference.

---

### Official Review · Reviewer_DB6a · 2025-02-27
**The proposed framework for Trustworthy AI is promising but needs stronger experimental validation and practical integration**

**Rating:** 6
**Confidence:** 3

**Review:**

The paper presents a framework for enhancing the trustworthiness of large language models (LLMs) through the use of polynomial-time (p-computable) checkers. This approach is motivated by the need for reliable AI, particularly in digital twin systems for smart cities, where LLMs are currently underutilized due to concerns over computational efficiency and hallucination risks. The authors propose a verification methodology based on the principle that solution verification is generally more computationally efficient than solution generation itself. The framework is evaluated using the 2-SAT problem, demonstrating its ability to correctly validate LLM-generated solutions while maintaining polynomial complexity.


## Strengths

- The paper addresses a relevant and important problem: ensuring trustworthiness in AI-generated solutions, particularly in resource-constrained environments such as digital twins.
- The proposed framework has a well-defined theoretical foundation, leveraging the distinction between problem-solving complexity and verification complexity to improve efficiency.
- The experiments, although limited in scope, provide an initial validation of the proposed concept.


## Weaknesses

- The scope of the experimental evaluation is somewhat narrow, as it is primarily demonstrated on the 2-SAT problem. A broader set of test cases or real-world applications could strengthen the claims about practical applicability.
- The explanation of the connection between LLM-generated solutions and formal logic checkers could be more detailed, particularly in terms of how errors are identified and corrected.
- There are minor typographical and formatting issues, including missing details in references, which should be addressed before final publication.


## Recommendation

Overall, the paper presents a novel and interesting approach to improving AI trustworthiness through polynomial-time checkers. While the concept is promising, additional experimental validation and a clearer discussion of practical applicability would improve its impact. Given its strengths and areas for improvement, I recommend a rating of 6: Marginally above acceptance threshold, with revisions to enhance clarity and applicability.

---

### Official Review · Reviewer_NBap · 2025-02-27
**Conceptual Framework for Trustworthy Artificial Intelligence: Combining Large Language Models with Formal Logic Systems**

**Rating:** 8
**Confidence:** 4

**Review:**

The paper tackles crucial issue: guaranteeing the trustworthiness of AI-generated solution. The growing adoption of AI is compounding concerns about the reliability of solutions derived from its usage, particularly regarding the application of large language models (LLMs).
A wider range of test scenarios or actual use cases would reinforce the assertions regarding practical feasibility. Despite these minor flaws, the article holds substantial theoretical and applied value.

---

### Official Review · Reviewer_heim · 2025-02-27
**The conceptual solution for verifying llm answers. Accept**

**Rating:** 8
**Confidence:** 3

**Review:**

The paper proposes a conceptual solution for verifying answers provided by large language models, especially in the context of application in digital twin systems for smart cities.
A note about the fact that the research includes testing only on certain tasks, will this limit the application of the results to other types of problems and use cases?

---

### Decision · Program_Chairs · 2025-03-08

**Decision:**

Accept (Oral)

**Comment:**

Your article has been accepted and you can give a talk on the article. All articles will be sorted by rating and within the available conference places one author from each article will be invited. If there are not enough places, then you will either have the opportunity to speak remotely or come at your own expense!